# Preparation of Biomimetic 3D Gastric Model with Photo-Curing Resin and Evaluation the Growth of *Helicobacter pylori*

**DOI:** 10.3390/polym13203593

**Published:** 2021-10-19

**Authors:** Yu-Tung Hsu, Ming-Hu Ho, Shiao-Pieng Lee, Chen-Yu Kao

**Affiliations:** 1Department of Chemical Engineering, National Taiwan University of Science and Technology, Taipei 10607, Taiwan; cindy010253@gmail.com (Y.-T.H.); mhho@mail.ntust.edu.tw (M.-H.H.); 2R&D Center for Membrane Technology, National Taiwan University of Science and Technology, Taipei 10607, Taiwan; 3Department of Dentistry, Division of Oral and Maxillofacial Surgery, Tri-Service General Hospital, Taipei 11490, Taiwan; shiao-pieng@yahoo.com.tw; 4Department of Biomedical Engineering, National Defense Medical Center, Taipei 11490, Taiwan; 5Graduate Institute of Biomedical Engineering, National Taiwan University of Science and Technology, Taipei 10607, Taiwan; 6Biomedical Engineering Research Center, National Defense Medical Center, Taipei 11490, Taiwan

**Keywords:** photo-curing 3D printing, resin, *cis*-1,4 polyisoprene, gastric model, *H. pylori* in vitro test

## Abstract

Three-dimensional (3D) printing technology is now widely used in biomedical developments. Especially, photo-curing systems provide high resolution and precision. The current goal of biomedical 3D printing technology is the printing of human organs, but the current commercial photo-curable materials generally have high mechanical strength that cannot meet the mechanical properties of the object to be printed. In this research, a gastric model was printed using a photo-curing 3D printing technique. To mimic the wrinkle pattern of human gastric tissue, *cis*-1,4 polyisoprene with different reactive diluents was mixed and identified a formulation that produced a print with human gastric softness. This research discussed the effect of the Young’s modulus of the material and elucidated the relationship between the degree of conversion rate and viscosity. After modifying the *cis*-1,4 polyisoprene surface from hydrophobic to hydrophilic, we then evaluated its adhesion efficiency for gastric mucin and the gastrointestinal-inhabiting bacterium *Helicobacter pylori*.

## 1. Introduction

In recent years, three-dimensional (3D) technology has been widely used in biomedical fields, such as biotechnology and organ construction (human organ transplantation) [1,2]. The American Society for Testing Materials (ASTM) defines 3D printing technology as “a process of joining materials” [3]. Applying a layered stacking technique, 3D printing technology converts a two-dimensional (2D) cross-section into a 3D structure. The 3D model data are processed by computer-aided design software, and the material layers are laminated and printed to produce the target object. In this way, objects with any complex shapes or porous structures can be manufactured [4,5].

According to a World Health Organization survey, nearly 120,000 organs are transplanted each year [6]. Maturing the technology of 3D organ printing would not only alleviate patient’s pain and waiting times but would also drastically change the medicines used in the medical industry [7].

The physical and mechanical properties of the cured product in a photo-curing system depend on the choice of photo-curing resin. The rheological properties of liquid resins are commonly adjusted by adding small molecule solvents or a diluent monomer. Moreover, the ratio of oligomer to monomer in a resin can be adjusted to control the mechanical properties of the material after photopolymerization. However, commercial photo-curable materials are limited by insufficient mechanical strength and elasticity, high price, and poor biocompatibility. Therefore, polymer resins with mechanical properties that match those of the object to be printed are urgently required.

In the past, the challenge faced by photo-curing printing was that finished products were fragile and have poor toughness. The main reason is the low viscosity of the material, which leads to a high degree of cross-linking. On the contrary, the high viscosity resin is difficult to print relatively [8]. In photo-curing printing technology, most of the literature focuses on the development of materials with strong and tough mechanical properties [9,10], and the previous study used ABS for stomach model printing. Those methods were lack of soft and biomimetic properties [11]. Therefore, this study used *cis*-polyisoprene and three different dilute monomers to print a biomimetic 3D gastric model with high flexibility properties, breaking through the lack of flexibility and gastric rugae in 3D gastric printing in the past.

In the current experiment, a human-stomach model with gastric rugae was created using photo-curing 3D printing technology. The mechanical properties of the stomach model were rendered biomimetic by a photo-resin based on *cis*-1,4 polyisoprene (IR). The IR resin was formulated from three reactive solvents: industry isobornyl acrylate (IBOA), cyclic trimethylolpropane formal acrylate (CTFA), and 2-(2-ethoxyethoxy) ethyl acrylate (EOEOEA). The 3D printing process ensures high consistency of each batch of experimental samples, avoids damage to living samples during experiments, and increases the diversity of results. This method also accords with humanitarianism ideals and avoids complicated human experimentation [12]. The influence of the gastric simulation sample on the tensile properties, degree of conversion, and viscosity of the material during the experiment are also discussed. The hydrophilicity of the intrinsically hydrophobic IR is improved by modifying the material surface, and its adhesion efficiency for gastric mucin and *Helicobacter pylori* are also evaluated.

## 2. Materials and Methods

*Cis-*1,4 Polyisoprene (97%), mucin from porcine stomach, urea and 1N hydrochloric acid were purchased from Sigma-Aldrich (St. Louis, MO, USA). 2-(2-Ethoxyethoxy) Ethyl Acrylate (EOEOEA), Isobornyl acrylate (IBOA), and CTFA were purchased from Qualipoly Chemical Corporation (Kaohsiung, Taiwan). 2,4,6-Trimethyl Benzoyl Diphenyl Phosphine Oxide (TPO, 98%) was purchased from Double Bond Chemical (New Taipei City, Taiwan). Toluene (99.5%) was purchased from J.T. Baker (Phillipsburg, NJ, USA). A BD GasPak^TM^ company container system, anaerobic jar, and anaerobe blood agar was purchased from Becton, Dickinson and Company (Franklin Lakes, NJ, USA). Acrylonitrile Butadiene Styrene filament were purchased from EXTEK Company (Taoyuan, Taiwan). Distilled deionized (DI) water was used throughout the experiment.

### 2.1. Preparation of Photo-Curing Resin with IR

IR were mixed with three different diluent monomers, IBOA, CTFA, and EOEOEA, at four ratios (3:1, 4:1, 5:1, 6:1), respectively. The resins were evenly mixed at room temperature at 100 rpm for 1 day. After adding 3% thermoplastic polyolefin (TPO), the resins were stirred at room temperature at 200 rpm. Finally, they were placed in a low-temperature vacuum-drying chamber to remove the bubbles created by the stirring and then stored in the dark at room temperature until used.

### 2.2. Viscosity Measurements of the Photo-Curing Resin

The viscosities of the resins were measured using a Brookfield DV-1 viscosity measurement system (Brookfield, MA, USA). This device can be connected to a circulating water tank for accurate temperature control of the sample. As the sample amount was very small (only 2.0 mL), its temperature quickly became constant. This water bath controls the temperature with an accuracy of ±0.05 °C, and its temperature range is −95–200 °C. For constant-temperature control, the constant-temperature water bath pump was driven into the outer flow channel in the viscometer sample tray. The spindle type was CPA-51Z. The shear rate (s^−1^) was calculated as the shear rate coefficient × the rotation speed (rpm), where the shear rate coefficient was *N* = 3.84. The measured viscosity range was 4.8–16,180 cP.

### 2.3. Stretch Properties of the Photo-Curing Resin

The printing process in this experiment used a digital light processing (DLP) system. As the layer-to-layer stacking was quite close and meticulous, the mechanical properties after direct molding were expected to remain nearly constant. In this study, silicon-rubber mold technology was applied to produce a flexible silicone mold with a certain dimensional accuracy, which ensured the integrity and accuracy of the sample. The stretching mold was cured using a DLP digital projector (H6510BD, Taipei, Taiwan) light source with a main emission wavelength of 405 nm, energy of 200 mW/cm^2^, working distance of 25 cm, and TPO light initiator.

The photopolymerization material was largely shrunk during the irradiation process. When the material was poured into the cavity of the silicon-rubber mold, its top layer was covered with a glass sheet. Once the material had polymerized and shrunk, it was removed and then cured under ultraviolet (UV) light (EB-UFL-40F1) for 20 min to obtain a tensile test sample that conformed to the standard. After forming the material into the required shape, the sample was stretched to breakage in a universal testing machine (M550-25AT) operated at 10 mm/min. The tensile test was performed in accordance with the material standard ASTM D-638-IV.

### 2.4. Fourier Transform Infrared Spectrometer Analysis

Fourier transform infrared spectroscopy (FTIR) was conducted in the mid-infrared range (4000–400 cm^−1^). The instrumental model was an FTS-3500 (BIO-RAD, Hercules, Cal, USA), and the sample was prepared as follows. First, the KBr pellet was uniformly spread with a certain amount of photo-curing resin daub. The photopolymerization degrees of the different photo-curable resins can be determined from the area changes of the characteristic absorption peaks of specific functional groups in the FTIR spectra. The signals of some common functional groups are the stretching vibration of C = C at 1638 cm^−1^, the bending vibration of = CH_2_ at 1410 cm^−1^, and the = CH bending vibration at 809 cm^−1^. In the current experiment, the changed area of the elastic vibration peak of C = C with a wavelength of 1638 cm^−1^ was measured, and we calculated the conversion rate from the detected ratio of C = C transfer to -C–C- as follows:(1)$$\mathrm{Conversion}\;(\%)=[1-\frac{A_{t}}{A_{0}}]\times 100\%$$
A_t_ is the integral of the area under the specific peak after t seconds of the photo-curable resin, and A_0_ is the area under the specific peak before the initial photo-curable resin has been exposed.

### 2.5. Resin Printing Formability

After measuring the Young’s moduli of the samples, the resin mixing ratio yielding the lowest Young’s modulus for 3D printing was selected because this parameter best matched that of the human gastric material. The most important process of 3D printing is the printing time of the first few layers. In this study, the total printing layer was 168 layers, and the printing time of the first two layers was set for 480 s. The remaining 166 layers were printed for 420 s in each layer. The total printing time was 19 h and 20 min.

### 2.6. Surface Modification of the Photo-Curing Resin

The resin sample was placed on a movable Atmospheric Pressure Plasma Jet (APPJ) platform. During the experiment, the sample was scanned back and forth by moving only along the XY axis. Each back-and-forth motion of the platform was regarded as one scan, and the typical number of scans per sample was five. The APPJ was set perpendicularly (90°), and the samples were set 2 mm from the nozzle. The APPJ controlled the back-and-forth scan movements of the platform. The sample was plasma-treated in a sealed acrylic box to avoid interference from external humidity and other airflows. The plasma power was 70 W/cm^3^_,_ and the working gas was high-purity Ar flowing at 3 slm. After scanning, the hydrophilicity change of the sample was observed by measuring its water contact angle (WCA).

### 2.7. WCA

The WCA for determining the hydrophilicity of the resin sample was measured as follows. Each sample was placed on the sample stage in air, and a constant volume of deionized water was dropped onto its surface. From an image of the sample, the contact angle between the water droplet and sample surface was calculated by computer software. At least five droplets were applied to ensure accurate measurements.

### 2.8. In Vitro Test of H. pylori

The *H. pylori* strain (ATCC 43504) used in this experiment required approximately 10 working days of cultivation. Prior to each experiment, the laminar flow was sterilized by wiping with alcohol. The experimental steps for cultivating *H. pylori* are described below.

The *H. pylori* strain was thawed at room temperature, and 10 mL brucella broth (BB10) containing 10% fetal bovine serum was added to the *H. pylori* strain solution. The culture solution containing the bacteria was placed in an anaerobic tank and incubated at 37 °C. On the following day, the absorbance, at 600 nm (OD600), of the culture medium was measured using a UV–Visible spectrophotometer. The required amount of bacterial solution to be added to the subculture was calculated from the OD600 value.

### 2.9. Culturing H. pylori Growth under Different Conditions

The growth of *H. pylori* before and after the plasma treatment were evaluated under different treatments. During this experiment, the mucin concentration was optimized to determine the most suitable environment for *H. pylori* growth and to conduct the follow-up research. A mucin concentration of 5% provided the most suitable growth environment for *H. pylori*. The four experimental treatments are described below and summarized in Table 1.

In the first treatment method, the quantitative *H. pylori* bacterial solution (2.4 × 10^8^ CFU) was added to each gastric model and incubated overnight at 37 °C. On the next day, the model was serially diluted with sterile PBS (pH = 7.4). The dilutions were inoculated onto blood agar plates, which were covered and placed upside down in an anaerobic jar with one GasPak and a wet tissue paper for 1–2 days to observe the survival status of *H. pylori.* In the second treatment method, the 5% mucin solution prepared in sterile PBS (pH = 7.4) was added to the 3D gastric model and stored overnight at 4 °C. The next day, the non-adhered mucin solution was removed from the model, and *H. pylori* bacterial solution was added. In subsequent steps, the gastric model was treated as described for the first treatment method. In the third treatment method, 5% mucin and 40 mM urea prepared in sterile PBS (pH = 7.4) were added to the model and stored overnight at 4 °C. The next day, *H. pylori* bacterial solution was added to the gastric model, and the model was subsequently treated as described for the first treatment method. In the fourth treatment method, 5% mucin and 40 mM urea prepared in sterile PBS (pH = 7.4) were added to the gastric model and stored overnight at 4 °C. The quantitative *H. pylori* bacterial solution (2.4 × 10^8^ CFU) was then added to each gastric model and incubated overnight at 37 °C. The next day, 5% mucin solution was added to the upper layer, and the model was placed in an anaerobic jar with one GasPak and a wet tissue paper overnight at 37 °C. The following day, the model was removed and serially diluted with sterile PBS (pH = 7.4). The dilutions were inoculated on blood agar plates, which were subsequently covered and placed upside down in an anaerobic jar with one GasPak and a wet tissue paper for 1–2 days to observe the survival status of *H. pylori.*

### 2.10. Evaluation of the H. pylori Adhesion Effect

After culturing with *H. pylori*, the resin and ABS samples were fixed in a beaker and stirred continuously for 24 h at 50 rpm in a simulated gastric fluid environment (pH = 2). The simulated gastric fluid was prepared by deionized water and then adjusted pH to 2 with diluted hydrochloric acid. After 2, 4, 6, and 24 h, the cultured sample was removed and serially diluted with sterile PBS (pH = 7.4). The dilutions were inoculated onto blood agar plates, which were covered and placed upside down in an anaerobic jar with one GasPak and a wet tissue paper for 1–2 days to observe the survival status of *H. pylori.*

The plasma treatment was both included and excluded to evaluate the attachment of *H. pylori* in the two situations.

### 2.11. Evaluation of the Antibacterial Effect of the 2D and 3D Models

In this experiment, the hard material ABS and the soft material IR/EOEOEA samples with the best *H. pylori* adhesion were selected for the evaluation of the antibacterial effect of tetracycline. After the 2D and 3D gastric-model printing of the IR/EOEOEA resin, the resin model and the ABS 2D model with plasma treatment and urea addition were used to culture *H. pylori*. The samples were fixed in a beaker, and a 3-mg dosage of a tetracycline-free drug was added to the beaker and stirred continuously for 24 h at 50 rpm in a simulated gastric fluid environment (pH = 2). At 2, 4, 6, and 24 h, the model was taken out and serially diluted with sterile PBS (pH = 7.4). The dilutions were inoculated onto blood agar plates, and the plates were covered and placed upside down inside an anaerobic jar with 1 GasPak and a wet tissue paper for 1–2 days to observe the antibacterial effect on the 2D and 3D models.

### 2.12. Statistical Analysis

Statistical software (Minitab, Taipei, Taiwan) was used for data analysis. The viscosity, Young’s Modulus, water contact angles, and in vitro test of *H. pylori* was analyzed by one-way ANOVA. The ANOVA test was carried out at a 95% confidence level. The viscosity, Young’s Modulus, water contact angles, and all the experiments related to *H. pylori* were expressed as mean ± standard error.

## 3. Results and Discussion

### 3.1. Effect of Diluent Monomers and IR on the Resin Rheology

The biocompatibility of IR was identified in previous researches by culturing L929 mouse fibroblast cell [13]. The material composed with IR also shows excellent biocompatibility result in the clinical trial [14]. The high biocompatibility is one of the main reason for us to choose IR as the printing resin. Another reason that must be considered in 3D printing is the viscosity of the material. Viscosity critically affects the physical properties of materials and applications. As IR has a very high viscosity (35,000 cp) [15], it is unsuitable for use in photo-curing 3D printing. Therefore, to control the viscosity for 3D printing applications, a photo-resin containing IR was designed and formulated.

Figure 1 shows the viscosities of the resins containing various proportions of IR and diluent monomers. The resin viscosities did not significantly depend on the mixing ratio of diluent monomer and IR but increased with increasing amounts of IR. The resins containing IBOA and CTFA (Figure 1a,b, respectively) were more viscous than the resin containing EOEOEA (Figure 1c). The ANOVA test indicated that there is no significant difference caused by the different ratios of IR/IBOA, IR/CTFA, and IR/EOEOEA. This behavior is possibly explained by the structure and hydrogen bonding of the diluent monomers. The steric hindrance caused by the nonlinear structures of IBOA and CTFA reduced the mobility of oligomers, thus decreasing the free volume in the photo-resin and increasing the viscosity. On the contrary, EOEOEA is linear with no steric hindrance, so it can decrease the viscosity of resin.

Oligomers with higher molecular weights have longer molecular chains, which induce higher Van der Waals forces between the molecules. The severe stacking and entanglement between the molecules raise the viscosity [16]. The viscosities of the three diluents, thus, decreased in the order of CTFA > IBOA > EOEOEA. At room temperature, the viscosity of EOEOEA was only 3–8 cp, revealing a low degree of intermolecular entanglement. On the contrary, both CTFA and IBOA exhibited high viscosities (6–10 and 15–20 cp, respectively, at room temperature). In the resin formulated with CTFA, the viscosity was increased by hydrogen bonding between the oligomer and IR, which was stronger for CTFA than the other reactive solvents [17].

### 3.2. Analysis of the Mechanical Properties of the Materials

The Young’s modulus of the fully photo-cured IR was 0.355 MPa [18], close to the mechanical properties of human gastric tissue [19]. However, IR is difficult to print and must be formulated to reduce its viscosity. To identify the material that best matches the mechanical properties of the human stomach, the Young’s moduli of cured resins with different formulations has been analyzed.

As shown in Figure 2a–c, the Young’s modulus increased with an increasing addition of IBOA and decreased with an increasing addition of CTFA and EOEOEA. The ANOVA test indicated that there has significant difference in IR/IBOA and IR/EOEOEA (*p* < 0.05 and *p* < 0.01). The increasing trend for IBOA was attributable to the hardness of the IBOA monomer (*T*g = 88 °C) in the resin, which enhanced the stiffness of the cured polymer. The soft reactive solvents with lower *T*g (27 °C and −53 °C for CTFA and EOEOEA, respectively) reduced the Young’s moduli of the corresponding resins. At temperatures below *T*g, the molecular chains lack sufficient energy for movement, and the materials exist in a rigid glass state. Contrarily, when the temperature is above *T*g, the polymeric chains are highly mobile in the amorphous state, and the softness and flexibility of the cured materials are similar to those of rubber [20].

### 3.3. Resin Printability

As shown in Figure 2g–i, the resin with the IR:EOEOEA = 3:1 formulation presented the lowest Young’s modulus among the formulations and high printability. The IR:EOEOEA = 6:1, IR:IBOA = 6:1, and IR:CTFA = 3:1 formulation also showed acceptable printability. In the following experiments, the resin with the lowest Young’s modulus was chosen as the gastric model because its mechanical properties resembled those of human gastric tissues [19].

### 3.4. Conversion Rates of the Photo-Curing Resins

As shown in Figure 3, the conversion rate was low in the IR resins containing small amounts of the dilute monomers IBOA, CTFA, and EOEOEA. Clearly, adding dilute monomers promoted the conversion rate of the studied photo-resin. Comparing the conversion rates with the viscosities (Figure 1), the dilute monomers simultaneously increased the conversion degree and decreased the viscosity of the photo-resin. The EOEOEA monomer minimized the viscosity and maximized the conversion rate, whereas CTFA induced the opposite behaviors.

The conversion degree was mainly related to the extent of intermolecular interactions, which was proportional to the crosslinking density. Low-viscosity resins form a highly cross-linked network because the mass-transfer resistance in the resin is low. Such high crosslinking density promotes the stability of cured resin. By contrast, high-viscosity resins hinder the mobility of oligomers and monomers, decreasing the conversion rate of the photo-curing resins. Theoretically, a high degree of crosslinking should toughen the polymers, but the low toughness of the resin containing EOEOEA indicated that the mechanical properties of the photo-resins used in this study were dominated by the *Tg* of the monomers and reactive solvents.

### 3.5. Resin-Surface Modification

The high hydrophobicity of IR prohibits bacterial attachment. To improve the hydrophilicity, the IR surface was modified by APPJ, the WCA and bacterial culturing on the surface were analyzed.

After treating the resin with APPJ in argon gas, the surface hydrophilicity was significantly improved (Figure 4). There has significant difference in IR: IBOA = 6:1 (*p* < 0.01). ABS has the best hydrophilicity after plasma treatment. Comparing the hydrophilicity of the two resin samples shows that IR/EOEOEA has better hydrophilicity. Plasma treatment at 70 W greatly improved the hydrophilicity by exciting the high-energy electrons and metastable molecules on the surface. The water vapor molecules then generated a large number of free radicals for the formation of C-O and C = O [21,22] on the material surfaces.

### 3.6. In Vitro Test of Helicobacter pylori

#### 3.6.1. Comparison of *H. pylori* Growth before and after Plasma Treatment

In this experiment, the growth of *H. pylori* on the resin and ABS samples was evaluated after the four treatments described in Table 1: no treatment, a one-layer mucin treatment, a one-layer mucin coating and one-layer urea coating (mucin/urea) treatment, and a one-layer mucin coating, one-layer urea coating, and one-layer mucin coating (mucin/urea) treatment.

The second treatment provided a single protection layer for *H. pylori*. The third treatment additionally provided urea, a known chemoattractant of *H. pylori*. The fourth treatment provided urea and an additional protective layer [23].

The *H. pylori* adhesion was greatly improved by plasma treatment on the 3D printed IR/IBOA, IR/EOEOEA, and ABS models (Figure 5a,b,c, respectively). The adhesion rate was most improved on IR: EOEOEA = 6:1, with an *H. pylori* attachment 6.64 times higher than on the untreated surface. Figure 5 also reveals that, after the plasma treatment, the surface attachment of *H. pylori* decreased in the order of IR/EOEOEA > ABS > IR/IBOA. There has significant difference in mucin/urea coating of all formulations (*p* < 0.05). The difference between IR/EOEOEA and ABS were statistically significant in mucin/urea/mucin coating (*p* < 0.05). The result support that resin and ABS samples with and without plasma treatment actually influence the *H. pylori* attachment effect.

The surface effect is dominated by two parameters: the material hydrophilicity and the topographical cure, including the gastric rugae on the material. The bacterial attachment on all surfaces was improved by the enhanced hydrophilicity conferred by APPJ. However, although ABS is more hydrophilic than IR/EOEOEA, the *H. pylori* attachment was much higher on IR/EOEOEA than on ABS. This result supports that bacterial attachment was further facilitated by the simulated gastric rugae and soft properties of IR/EOEOEA.

The *H. pylori* adhesion was promoted by the mucin and urea coatings, and it increased with number of coating layers. Mucin and urea are naturally found on human gastric walls. Reportedly, mucin and urea form a protective layer for *H. pylori*, and urea is hydrolyzed by urease to produce ammonia and neutralize the gastric acid, providing a neutral microenvironment for *H. pylori* survival [23,24]. Thus, the mucin and urea coating can be reasonably assumed to promote the growth of *H. pylori*. The results revealed that our 3D-printed stomach model is highly biomimetic of the real human stomach.

#### 3.6.2. Culturing *H. pylori* in a pH = 2 Environment

In this experiment, the photo-cured surfaces cultured with *H. pylori* were immersed into a pH 2 buffer for one day, which simulates the in vivo environment. As shown in Figure 6, *H. pylori* did not survive the pH 2 solution without the mucin and urea coating. On the gastric mucin coating, the stability of bacterial attachment was improved but remained much lower than under the in vivo condition. The bacterial adhesion was highest on the samples with the mucin/urea coating. This high survival rate of *H. pylori* in the pH 2 environment supports that the coating layer effectively protects the *H. pylori* cells. More importantly, the survival rate of *H. pylori* was much higher on the IR/EOEOEA photo-resin than on IR/IBOA and ABS. The result reveals that, besides possessing biomimetic mechanical properties, IR/EOEOEA replicated the biological characteristics of the host tissue.

#### 3.6.3. Comparison of *H. pylori* Adhesion Effect

In this experiment, *H. pylori* was cultured on IR/IBOA, IR/EOEOEA, and ABS for 2, 4, 6, and 24 h with shaking at 50 rpm in the pH 2 environment. Some of the cells were detached in the shaking incubator, and the remaining *H. pylori* were quantified as in Figure 6. Both with and without the plasma treatment (Figure 7a,b, respectively), the adhesion of *H. pylori* was lowest on ABS, the most common material used in stomach models, and was highest on IR/EOEOEA, consistent with the results of Figure 6. Although the bacterial growths were scarcely higher on IR/EOEOEA and IR/IBOA than on ABS under static conditions with pH = 7, the durations of the attachment were significantly higher than on ABS, speculating that the irregular 3D-printed pattern and flexible mechanical characteristics of the IR resin efficiently protected *H. pylori* from the fluid flow and acidic environment. Comparing the attachments on IR/IBOA and IR/EOEOEA in Figure 7, more bacterial colonies on the latter surface was observed, implying that the soft and flexible surfaces can promote the survival of *H. pylori* under dynamic and acidic conditions.

### 3.7. Evaluation of Antibacterial Effect in the 2D and 3D Models

Previous studies have shown that the antibacterial effect of resins is mostly presented in 2D models [24]. In this study, *H. pylori* was cultured on ABS and 2D and 3D samples of IR/EOEOEA and continuously stirred for 2, 4, 6, and 24 h at 50 rpm in a pH 2 environment. The samples were then dosed with a tetracycline-free drug. The results showed that the tetracycline-free drug had the best antibacterial effect on the IR/EOEOEA 2D model, as shown in Figure 8 The IR/EOEOEA 3D model had the lowest antibacterial effect. The antibacterial properties of the tetracycline-free drug on *H. pylori* affected the samples over 24 h. After 4 h of administration, the amount of *H. pylori* attached to the IR/EOEOEA 3D model was 1.96 times higher than the 2D model, and the attachment amount of the 3D model was 1.37 times higher than the ABS.

Figure 9 shows the adhesion effect of *H. pylori* without dosed tetracycline-free drug and the antibacterial effect of *H. pylori* after dosed tetracycline of three different models. It can be clearly seen that the model with the best adhesion effect was 3D IR/EOEOEA (Figure 9b). After treatment with tetracycline for 24 h, there are still a few amounts of *H. pylori* attached on the 3D IR/EOEOEA model.

According to a previous report, *H. pylori* cannot be completely eradicated by tetracycline; it can only achieve an inhibitory effect [25]. The results of this study’s in vitro experiments indicated that that, if tetracycline is administered on the same day, the inhibition of *H. pylori* by tetracycline can only achieve a slight, but reproducible, reduction [26].

Comparing the results of the 2D and 3D models prepared for this experiment, the 3D gastric model simulated gastric rugae, which can promote and protect the growth of *H. pylori* and had biomimetic properties. The results of the *H. pylori* attachment to the 3D model were similar to results of in vivo tests published in the literature, which inhibited *H. pylori* but did not entirely eradicate it.

## 4. Conclusions

Using 3D printing technology, this research fabricated a gastric model that mimicked the mechanical strength and surface chemistry of the human stomach. A novel photo-curable resin was developed, and its printability and physical properties were characterized. The viscosity increased with the increasing weight percentage of IR, and the conversion degree of photo-curing increased with the amount of reactive diluent solvent. Plasma treatment significantly improved the surface hydrophilicity of the printed model. In an antibacterial test using 3 mg of tetracycline, the antibacterial effect was high for the 2D model and weak for the 3D model. In the *H. pylori* culture experiments, *H. pylori* grew more vigorously on the photo-cured 3D printed model than on the ABS, indicating that the printed model developed here was a more suitable gastric model with better biomimetic properties than ABS, the standard material used for gastric models.

## Figures and Tables

**Figure 1 polymers-13-03593-f001:**
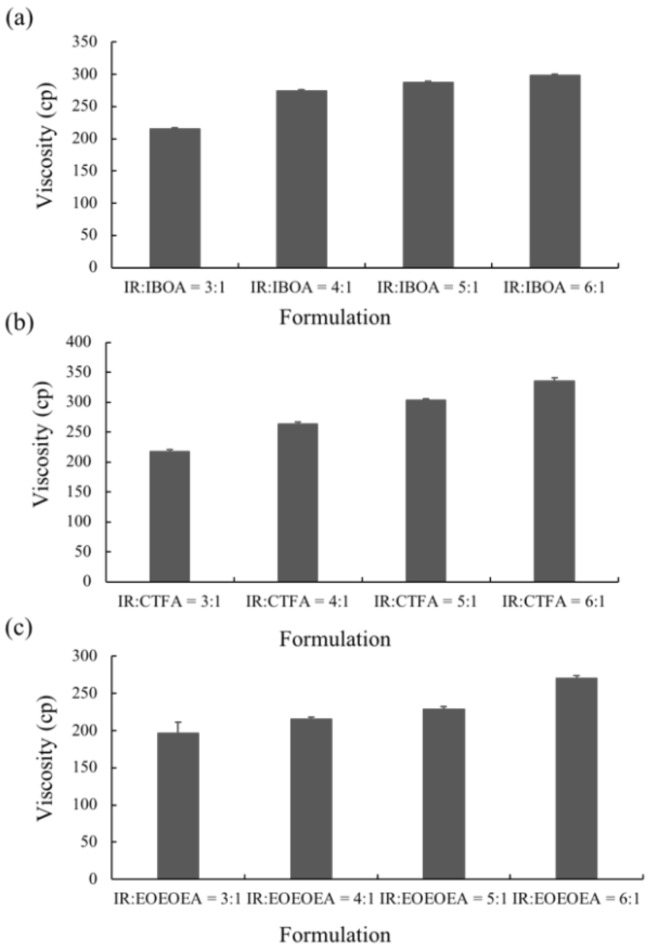
The viscosity of various proportions of *cis*-1,4 polyisoprene (IR) and diluent monomer formulations. (**a**) IR/IBOA, (**b**) IR/CTFA, (**c**) IR/EOEOEA. The ratios of polyisoprene (IR) and diluent monomers are 3:1, 4:1, 5:1, and 6:1.

**Figure 2 polymers-13-03593-f002:**
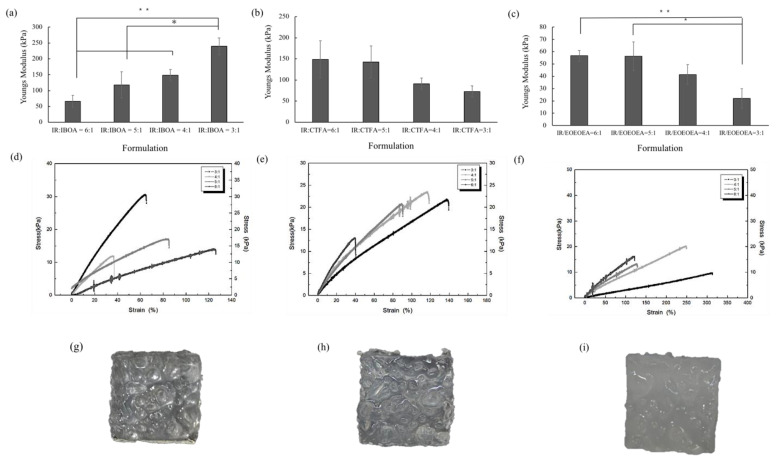
The Young’s modulus, stress-strain curves and 3D printing schematic of stomach simulation of *cis*-1,4 polyisoprene (IR) and three diluent monomers. (**a**–**c**) Comparing the mechanical properties of polyisoprene mixed with IBOA, CTFA, and EOEOEA in different proportions ((**a**) IR/IBOA, (**b**) IR/CTFA, (**c**) IR/EOEOEA). The significant difference between resin formulations was indicated by * (*p* < 0.05) and ** (*p* < 0.01). (**d**–**f**) The stress-strain curves of IR and three diluent monomers ((**d**) IR/IBOA, (**e**) IR/CTFA, (**f**) IR/EOEOEA). (**g**–**i**) Three-dimensional printing schematic of stomach simulation (**g**) IR/IBOA, (**h**) IR/CTFA, (**i**) IR/EOEOEA.

**Figure 3 polymers-13-03593-f003:**
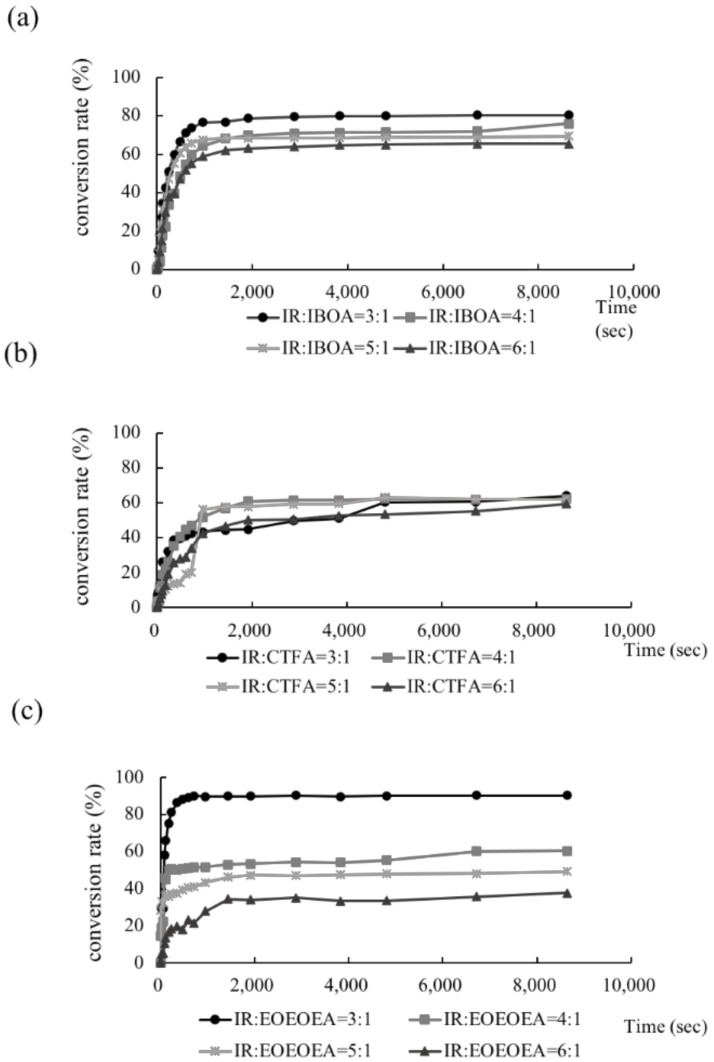
Degree of Conversion rate of *cis*-polyisoprene (IR) with different dilute monomers. (**a**) IR/IBOA, (**b**) IR/CTFA, (**c**) IR/EOEOEA. The ratios of polyisoprene (IR) and dilute monomers are 3:1, 4:1, 5:1, and 6:1.

**Figure 4 polymers-13-03593-f004:**
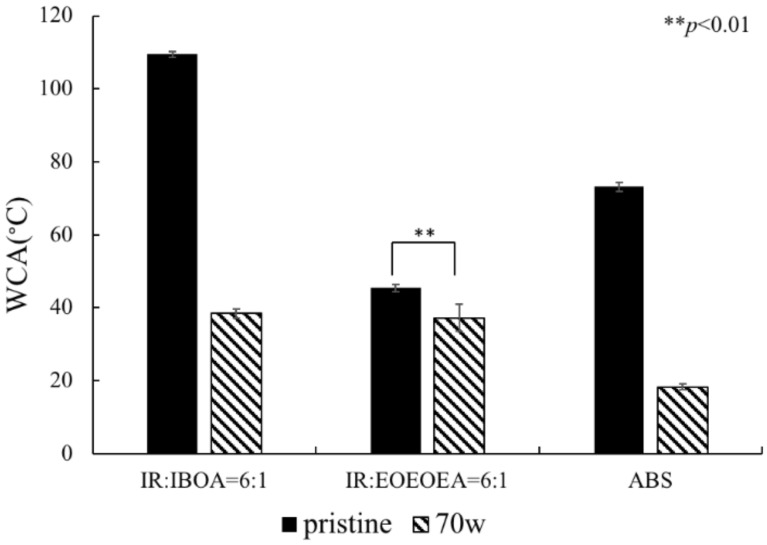
Contact angles of IR: IBOA = 6:1, IR: IEOEOEA = 6:1, and ABS. The power of plasma was 70 W/cm^3^, and the high-purity Ar with a flow rate of 3 slm is used as the working gas. (*) indicates a significant difference from each group.

**Figure 5 polymers-13-03593-f005:**
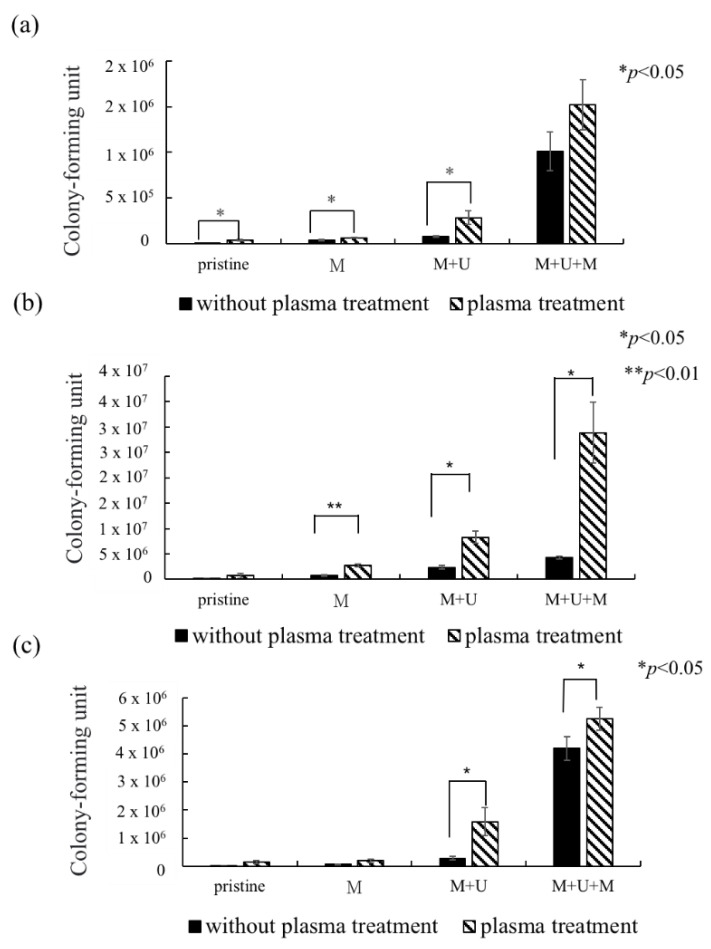
The growth of *H. pylori* on photo-cured resin and ABS after culture for 24 h. (**a**) IR:IBOA = 6:1, (**b**) IR:EOEOEA = 6:1, (**c**) ABS. M means mucin coating, M + U means mucin/urea coating, and M + U + M means mucin/urea/mucin coating. (*) indicates a significant difference from each group.

**Figure 6 polymers-13-03593-f006:**
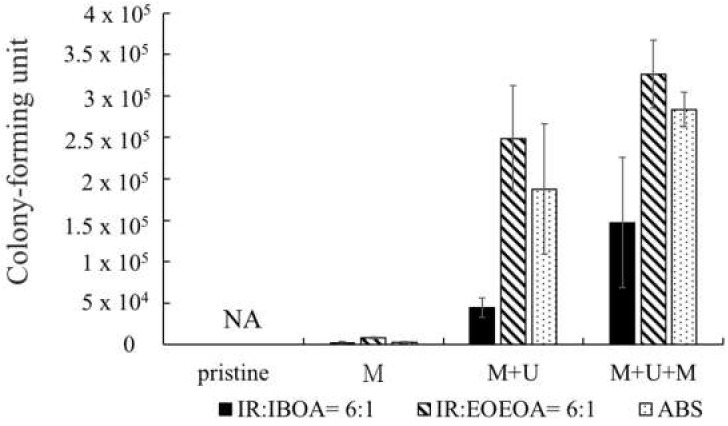
Different method of culturing *H. pylori* on resin and ABS under the pH 2 environment. M means mucin coating, M + U means mucin/urea coating, and M + U + M means mucin/urea/mucin coating. All the surfaces were treated by plasma before mucin and urea coating.

**Figure 7 polymers-13-03593-f007:**
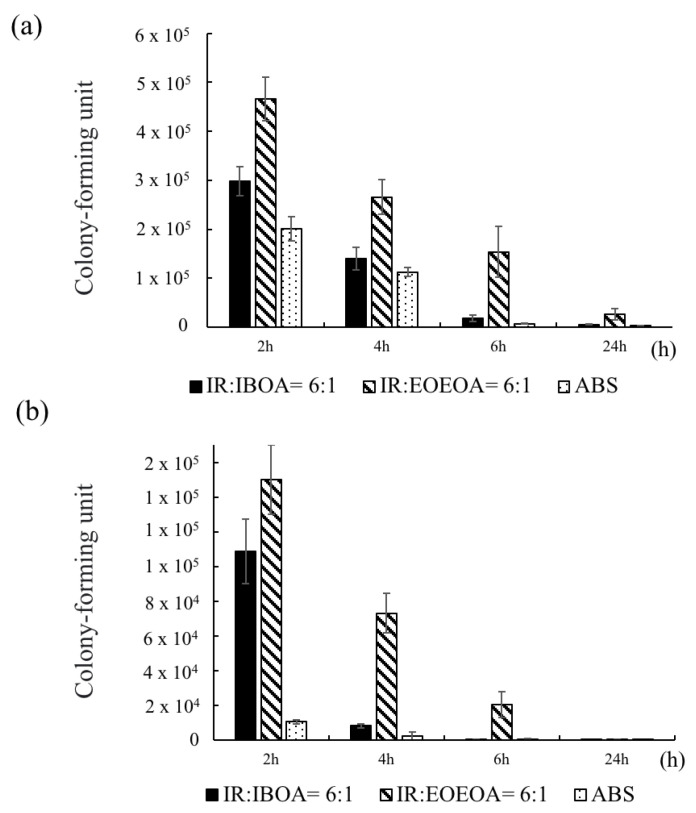
The *H. pylori* adhesion effect with and without plasma treatment in different time under a shaking of 50 rpm in the pH 2 environment. (**a**) *H. pylori* adhesion effect on resin and ABS with plasma treatment. (**b**) *H. pylori* adhesion effect on resin and ABS without plasma treatment.

**Figure 8 polymers-13-03593-f008:**
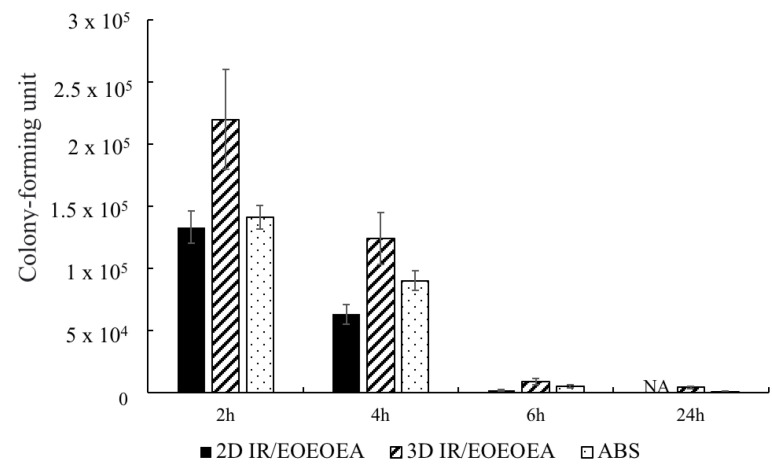
The anti-bacterial effect of 2D and 3D model in different time under a shaking of 50 rpm in the pH 2 environment. Two-dimensional IR/EOEOEA is a photo-cured flat surface sample. Three-dimensional IR/EOEOEA is a photo-cured sample with gastric rugae. ABS is a flat surface sample.

**Figure 9 polymers-13-03593-f009:**
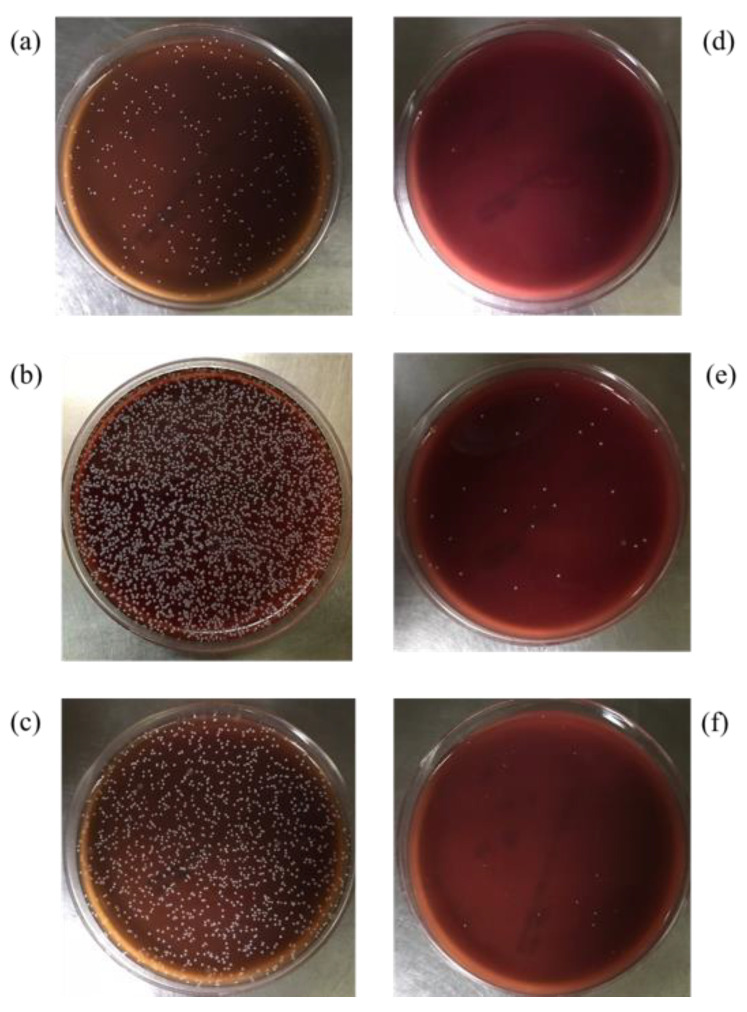
The images of *H. pylori* adhesion effect and antibacterial effects. (**a**,**d**) Two-dimensional IR/EOEOEA model. (**b**,**e**) Three-dimensional IR/EOEOEA model. (**c**,**f**) ABS model. (**a**–**c**) Adhesion effect of *H. pylori* without tetracycline treatment. (**d**–**f**) The antibacterial effects of *H. pylori* with tetracycline treatment for 24 h.

**Table 1 polymers-13-03593-t001:** Different treatments for culturing *H. pylori*.

	Pristine	Coating mucin	Coating mucin and urea	**Coating Mucin and Urea** **Coating Mucin on the Top Layer**
Illustration	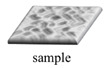	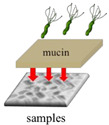	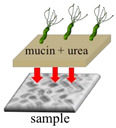	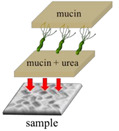
	**Pristine**	**Coating mucin**	**Coating mucin and urea**	**Coating mucin and urea** **Coating mucin on the top layer**
Illustration	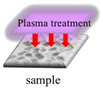	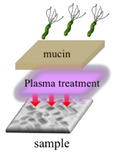	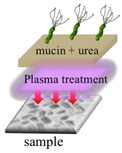	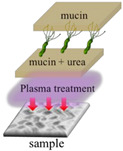

## Data Availability

The data presented in this study are available on request from the corresponding author.

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
