# Peer review of "Preparation of Biomimetic 3D Gastric Model with Photo-Curing Resin and Evaluation the Growth of Helicobacter pylori"

_polymers, 2021, doi:10.3390/polym13203593_

Round 1

Reviewer 1 Report

In this manuscript (polymers-1406932), authors have prepared biomimetic 3D gastric model with photo-curing resin and evaluated the growth of Helicobacter Pylori. This study is interesting and can be considered for publication in this journal after a major revision.

  1. Authors should provide a digital image of the printing process (including a schematic procedure) and a short video of the printing of the biomimetic 3D gastric models in the supporting information for clear understanding of the process for future research directions.
  2. For mechanical analysis, please provide stress-strain curves for all tested samples in addition to the plots in Figure 2.
  3. For figures 6-9, is it possible to provide digital images of H.Pylori adhesion effect and antibacterial effects?

This revision will make this article effective for future research directions in this area.

Author Response

We are very grateful to your comments for the manuscript. According with your advice, we amended the relevant parts in manuscript. By carrying out experiments, finishing statistical analysis, and polishing the discussion, we sincerely hope that this manuscript has been improved. All of your questions are answered below. The modified parts in our revised manuscript are marked as red.

Point 1. Authors should provide a digital image of the printing process (including a schematic procedure) and a short video of the printing of the biomimetic 3D gastric models in the supporting information for clear understanding of the process for future research directions.

Response 1: Thanks for your suggestion. The digital image has been added into the graphical abstract. The video was added into the supplement data. The printing situation such as total printing time and the printing time for each layer has been added into the revised manuscript in line 135-139, p.3. 

https://drive.google.com/file/d/1CULORRaWRfj2miQt_yjlp6eThu13kcZT/view?usp=sharing

Point 2. For mechanical analysis, please provide stress-strain curves for all tested samples in addition to the plots in Figure 2.

Response 2: Thank you for the suggestion. The stress-strain curves have been added to the manuscript as Figures 2 (d), (e) and (f). (Line 285, p.8)

Point 3. For figures 6-9, is it possible to provide digital images of H. pylori adhesion effect and antibacterial effects?

Response 3: Thank you for the suggestion. The digital images have been added in the revised paper as Figure 9 (p. 14). The description was added in our manuscript, too. (Lines 413-417, p. 13)

Reviewer 2 Report

This article is interesting and novel. Nevertheless, it is necessary some modifications before its publication:

1) The introduction must be improved adding an adequate discussion of the recent research and the novelty of this work respect the other ones already published.

2) You must ommit the plural first person in the manuscript.

3) The statistical analysis must be better explained.

4) The reference list is lack of novel references about the last three years. So, more recent references must be incorporated.

Author Response

Responses to reviewers

We are very grateful to your comments for the manuscript. According with your advice, we amended the relevant parts in manuscript. By carrying out experiments, finishing statistical analysis, and polishing the discussion, we sincerely hope that this manuscript has been improved. All of your questions are answered below. The modified parts in our revised manuscript are marked as red.

Response to Reviewer 2 Comments

Point 1. The introduction must be improved adding an adequate discussion of the recent research and the novelty of this work respect the other ones already published.

Response 1: Thank you for the suggestion. We have improved the discussion of previous applications on photo-curing systems. The novelty of this research has been also added to the revised manuscript. (Lines 50-59, p. 2)

Point 2. You must omit the plural first person in the manuscript.

Response 2: Thank you for the suggestion. The manuscript has been modified.

Point 3. The statistical analysis must be better explained.

Response 3: Thank you for the suggestion. The more describe of statistical were added in the manuscript in Lines 224-226, 241-242, 269-270, 319 and 344-346.

Point 4. The reference list is lack of novel references about the last three years. So, more recent references must be incorporated.

Response 4: Thank you for the suggestion. The novel references about the last three years has been added to the revised manuscript. (References 2, 7, 8, 9, 12, 15, 24)

Reviewer 3 Report

The manuscript is very interesting reporting on the development of a biomimetic 3D Gastric Model using DLP 3D printing and the assessment of its suitability for the growth if H. Pylori. Some minor corrections should be addressed as follows:

Title: Please rephrase the title to be grammatically correct  ‘..and Evaluate the Growth of Helicobacter Pylori’ e.g ‘..for the evaluation of the growth..’

Line 29 and line 371, 379: add a full stop

Table 1 may be omitted and the information can be provided in brackets within the text.

Line 87: do authors mean ‘digital light processing’ instead of ‘data loss prevention’?

line 108: rephrase ‘can be judged’

line 112: Please use the correct terminology for ‘the mass area’.

Merge figure 3 with figure 2.

Line 280: correct the following phrase ‘The ratios are all polyisoprene (IR) and dilute monomers’

Figure 5: correct the units for water contact angle in (o) and elaborate more on the results presented in the graph on the text.

Figure 6 and figure 8: increase the font size of x and y axis

Line 342 and 390: correct ‘pH2’

Line 362: correct ‘The effect of H. pylori adhesion effect with’

Line 380-382: remove the title of the reference from the text

Line 384: replace the word ‘hide’ with more appropriate terminology

Line 389: correct ‘The anti-bacteria effect’

No information is provided in the materials and methods for the composition or even the use of the buffer simulating the ‘pH 2 environment’.

What about the biocompatibility of the materials used in the current study? please discuss on that.

Also, provide more information in the introduction on previous systems used for such applications and what were the challenges encountered with those systems to motivate this study

Author Response

Responses to reviewers

We are very grateful to your comments for the manuscript. According with your advice, we amended the relevant parts in manuscript. By carrying out experiments, finishing statistical analysis, and polishing the discussion, we sincerely hope that this manuscript has been improved. All of your questions are answered below. The modified parts in our revised manuscript are marked as red.

Point 1. Title: Please rephrase the title to be grammatically correct ‘..and Evaluate the Growth of Helicobacter Pylori’ e.g ‘..for the evaluation of the growth..’

Response 1: Thank you for the mention. The title has been modified to “Preparation of Biomimetic 3D Gastric Model with Photo-Curing Resin and Evaluation the Growth of Helicobacter Pylori.”

Point 2. Line 29 and line 371, 379: add a full stop

Response 2. Thank you for the mention. It has been modified in the revised manuscript. (Line 32, p.1) (Line 407, p.13) (Lines 419-421, p.13)

Point 3. Table 1 may be omitted and the information can be provided in brackets within the text.

Response 3: Thank you for the suggestion. Table 1 has been removed from the revised manuscript. The information of resin mixing ratio has been provided in the text. (Line 84, p.2)

Point 4. Line 87: do authors mean ‘digital light processing’ instead of ‘data loss prevention’?

Response 4: Thank you for the mention. The description has been modified to digital light processing. (Line 101, p.3)

Point 5. line 108: rephrase ‘can be judged’

Response 5: Thank you for the suggestion. The description has been modified to “can be determined” (Line 122, p.3)

Point 6. line 112: Please use the correct terminology for ‘the mass area’.

Response 6: Thank you for the suggestion. The description has been modified to “the changed area of the elastic vibration peak “ in line 126, p.3.

Point 7. Merge figure 3 with figure 2.

Response 7: Thank you for the suggestion. Figure 2 and figure 3 has been merged to the revised manuscript. (p.8)

Point 8. Line 280: correct the following phrase ‘The ratios are all polyisoprene (IR) and dilute monomers’

Response 8: Thank you for the mention. We have modified the description to “The ratios of polyisoprene (IR) and dilute monomers are 3:1, 4:1, 5:1, and 6:1.” (Line 259, p.7)

Point 9. Figure 5: correct the units for water contact angle in (o) and elaborate more on the results presented in the graph on the text.

Response 9: Thank you for the mention. The figure has been modified. The elaborate has been added in the text in lines 319-321, p.9.

Point 10. Figure 6 and figure 8: increase the font size of x and y axis

Response 10: Thank you for the suggestion. The font size of x and y for the two figures were modified. (p. 12, 13)

Point 11. Line 342 and 390: correct ‘pH2’

Response 11: Thank you for the mention. The word has been corrected. The word has been modified to “pH 2” in the revised manuscript in line 380, p.12 and line 429, p.13.

Point 12. Line 362: correct ‘The effect of H. pylori adhesion effect with’

Response 12: Thank you for the suggestion. The description has been modified to “The H. pylori adhesion effect” in the revised manuscript in line 399, p.13.

Point 13. Line 380-382: remove the title of the reference from the text

Response 13: Thank you for the mention. The title of the reference has been removed. (Line 420, p.13)

Point 14. Line 384: replace the word ‘hide’ with more appropriate terminology

Response 14: Thank you for the suggestion. We have changed the word to “protect” in the revised manuscript in line 423, p.13.

Point 15. Line 389: correct ‘The anti-bacteria effect’

Response 15: Thank you for the suggestion. The description has been corrected to “The anti-bacterial effect” in the revised manuscript in line 428, p.13

Point 16. No information is provided in the materials and methods for the composition or even the use of the buffer simulating the ‘pH 2 environment’.

Response 16: Thank you for the suggestion. The information of pH 2 environment has been added in the material part in Lines 74-75. The simulation method has been added in lines 202-203, too.

Point 17. What about the biocompatibility of the materials used in the current study? please discuss on that.

Response 17: The biocompatibility of the sample was discussed in lines 229-234, p.6. We refer to the pervious study in biocompatibility of IR, and elaborate the reason why this study choosed IR as the material.

Point 18. Also, provide more information in the introduction on previous systems used for such applications and what were the challenges encountered with those systems to motivate this study

Response 18: Thank you for the suggestion. We have improved the described of previous applications on photo-curing system. The motivation of this study has been added and mentioned in the revised manuscript. (Lines 50-59, p. 2)

Round 2

Reviewer 2 Report

The authors have improved the manuscript following the reviewer' suggestions. So, I recommend its publication in the present form.